# It's Hard for Neural Networks to Learn the Game of Life

## Abstract

Efforts to improve the learning abilities of neural networks have focused mostly on the role of optimization methods rather than on weight initializations. Recent findings, however, suggest that neural networks rely on lucky random initial weights of subnetworks called "lottery tickets" that converge quickly to a solution (Frankle & Carbin, 2018). To investigate how weight initializations affect performance, we examine small convolutional networks that are trained to predict $n$ steps of the two-dimensional cellular automaton *Conway's Game of Life*, the update rules of which can be implemented efficiently in a small CNN. We find that networks of this architecture trained on this task rarely converge. Rather, networks require substantially more parameters to consistently converge. Furthermore, we find that the initialization parameters that gradient descent converges to a solution are sensitive to small perturbations, such as a single sign change. Finally, we observe a critical value $d_0$ such that training minimal networks with examples in which cells are alive with probability $d_0$ dramatically increases the chance of convergence to a solution. Our results are consistent with the lottery ticket hypothesis (Frankle & Carbin, 2018).

## 1 Introduction

Recent findings suggest that neural networks can be "pruned" by 90% or more to eliminate unnecessary weights while maintaining performance similar to the original network . Similarly, the *lottery ticket hypothesis* (Frankle & Carbin, 2018) proposes that neural networks contain subnetworks, called *winning tickets*, that can be trained in isolation to reach the performance of the original. These results suggest that neural networks may rely on these lucky initializations to learn a good solution. Rather than extensively exploring weight-space, networks trained with gradient-based optimizers may converge quickly to local minima that are nearby the initialization, many of which will be poor estimators of the dataset distribution. If some subset of the weights must be in a winning configuration for a neural network to learn a good solution to a problem, then neural networks initialized with random weights must be significantly larger than the minimal network configuration that would solve the problem in order to optimize the chance having a winning initialization. Furthermore, small networks with winning initial configurations may be sensitive to small perturbations.

Similarly, gradient-based optimizers sample the gradient of the loss function with respect to the weights by averaging the gradient at a few elements of the dataset. Thus, a biased training dataset may bias the gradient in a way that can be detrimental to the success of the network. Here we examine how the distribution of the training dataset affects the network's ability to learn.

In this paper, we explore how effectively small neural networks learn to take as input a configuration for Conway's Game of Life (*Life*), and then output the configuration $n$ steps in the future. Since this task can be implemented minimally in a convolutional neural network with $2n + 1$ layers and $23n + 2$ trainable parameters, a neural network with identical architecture should, in principle, be able to learn a similar solution. Nonetheless, we find that networks of this architecture rarely find solutions. We show that the number of weights necessary for networks to reliably converge on a solution increases quickly with $n$. Additionally, we show that the probability of convergence is highly sensitive to small perturbations of initial weights. Finally, we explore properties of the training data that significantly increase the probability that a network will converge to a correct solution. While Life is a toy problem, we believe that these studies give insight into more general issues with training

neural networks. In particular, we expect that other neural network architectures and problems exhibit similar issues. We expect that networks likely require a large number of parameters to learn any domain, and that small networks likely exhibit similar sensitivities to small perturbations to their weights. Furthermore, optimal training datasets may be highly particular to certain parameters. Thus, with the growing interest in efficient neural networks (Han et al., 2015; Hassibi & Stork, 1993; Hinton et al., 2015; LeCun et al., 1990; Li et al., 2016), this results serve as an important step forward in developing ideal training conditions.

## 1.1 CONWAY'S GAME OF LIFE

Prior studies have shown interest in applying neural networks to model physical phenomena in applications including weather simulation and fluid dynamics (Baboo & Shereef, 2010; Maqsood et al., 2004; Mohan & Gaitonde, 2018; Shrivastava et al., 2012). Similarly, neural networks are trained to learn computational tasks, such as adding and multiplying (Kaiser & Sutskever, 2015; Graves et al., 2014; Joulin & Mikolov, 2015; Trask et al., 2018). In all of these tasks, neural networks are required to learn hidden-step processes in which the network must learn some update rule that can be generalized to perform multi-step computation.

Conway's *Life* is a two-dimensional cellular automaton with a simple local update rule that can produce complex global behavior. In a Life configuration, cells in an $n \times m$ grid can be either alive or dead (represented by $1$ or $0$ respectively). To determine the state of a given cell on the next step, Life considers the $3 \times 3$ grid of neighbors around the cell. Every step, cells with exactly two alive neighbors will maintain their state, cells with exactly three alive neighbors will become alive, and cells with any other number of neighbors will die (Figure 1). We consider a variant of Life in which cells outside of the $n \times m$ grid are always considered to be dead. Despite the simplicity of the update rule, Life can produce complex output over time, and thus can serve as an idealized problem for modeling hidden-step behavior.

## 2 RELATED WORK

Convolutional models of cellular automata including the Game of Life have been studied by Gilpin (2019), who classifies structural representations of the learned solutions to different cellular automata. Furthermore, Gilpin notes that narrow networks do not often converge, and for consistent convergence behavior, the cellular networks must be sufficiently wide. Our work in this paper quantifies this result, and in addition explores the sensitivity of initial conditions to perturbations and different dataset distributions.

Prior research has shown interest in whether neural networks can learn particular tasks. Joulin & Mikolov (2015) argue that certain recurrent neural networks cannot learn addition in a way that generalizes to an arbitrary number of bits. Theoretical work has shown that sufficiently overparameterized neural networks converge to global minima (Oymak & Soltanolkotabi, 2020; Du et al., 2018). Further theoretical work has found methods to minimize local minima (Kawaguchi & Kaelbling, 2019; Nguyen & Hein, 2017; Kawaguchi, 2016). Nye & Saxe (2018) show that minimal networks for the parity function and fast Fourier transform do not converge to a solution unless they are initialized close to a solution.

Increasing the depth and number of parameters of neural networks has been shown to increase the speed at which networks converge and their testing performance (Arora et al., 2018; Park et al., 2019). Similarly, Frankle & Carbin (2018) find that increasing parameter count can increase the chance of convergence to a good solution. Similarly, Li et al. (2018) and Neyshabur et al. (2018) find that training near-minimal networks leads to poor performance. Choromanska et al. (2015) provide some theoretical insight into why small networks are more likely to find poor local minima.

Weight initialization has been shown to matter in training deep neural networks. Glorot & Bengio (2010) find that initial weights should be normalized with respect to the size of each layer. Dauphin & Schoenholz (2019) find that tuning weight norms prior to training can increase training performance. Similarly, Mishkin & Matas (2016) propose a method for finding a good weight initialization for learning. Zhou et al. (2020) find that the sign of initial weights can determine if a particular subnetwork will converge to a good solution.

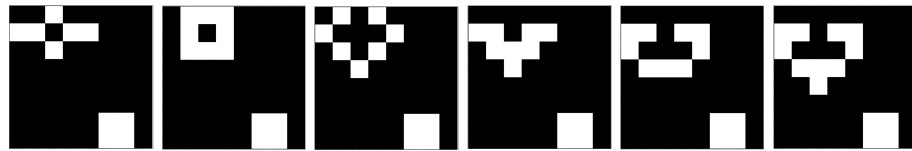

Figure 1: An example of an $8 \times 8$ cell board of Life over six time steps, evolving in time from left to right. White pixels are considered alive and black pixels are considered dead.

There is significant research into weight pruning and developing efficient networks (LeCun et al., 1990; Hassibi & Stork, 1993; Han et al., 2015; Li et al., 2016; Hinton et al., 2015; Li et al., 2018), including the lottery ticket hypothesis, which suggests that gradient descent allows lucky subnetworks to quickly converge to a solution (Frankle & Carbin, 2018).

Finally, there is interest in learning hidden step computational processes including algorithms and arithmetic (Kaiser & Sutskever, 2015; Graves et al., 2014; Joulin & Mikolov, 2015; Trask et al., 2018), fluid dynamics (Baboo & Shereef, 2010; Maqsood et al., 2004; Mohan & Gaitonde, 2018), and weather simulation (Shrivastava et al., 2012).

Among the above papers, there are studies that have already shown that weight initialization, over-parameterization, and training dataset statistics can determine whether or not a neural network can converge to a good solution to a problem. In this paper we build on these results to quantify them in the case of a simple problem with a known minimal solution. This allows us to derive empirical insights that may not be possible to observe where a perfect solution is not known, such as in the vision or language domains.

## 3 EXPERIMENTS AND RESULTS

We define the *Life problem* as a function-learning problem. In particular, if $x$ is a matrix of 1s and 0s, define $G(x)$ to be the next step in Life, according to the previously described update rules. Then, we define the Life problem to be the problem of predicting $G(x)$ given $x$. Similarly, we define the $n$-step-Life problem as the problem of learning to predict $G^n(x)$ given $x$. Since Life has a local update rule that considers a $3 \times 3$ grid to determine the state of the center cell, we can model Life with an entirely convolutional neural network, i.e., a neural network without any fully connected or pooling layers. A convolutional layer with two $3 \times 3$ filters that feeds into a second convolutional layer with one $1 \times 1$ filter, can solve the 1-step-Life problem efficiently, i.e., any fewer layers or convolutional filters would yield an architecture which cannot implement 1-step-Life. Thus, we call it the *minimal* architecture for Life. We use ReLU activation functions to prevent vanishing gradients for when the architecture is generalized to the $n$-step-Life problem by stacking layers. The second convolutional layer feeds into a final convolutional output layer with one $1 \times 1$ filter with a sigmoid activation function. This forces all outputs to approximate either 0 or 1 but does not, on its own, perform meaningful computation, and thus is included for this convenience. With appropriate weights, this constructs a three-layer convolutional neural network that can solve the 1-step-Life problem with 25 weights. We generalize this architecture to solve the $n$-step-Life problem by stacking $n$ copies of this network, as shown in Figure 2 (right).

We have hand-engineered weights for these architectures that implement the underlying rule and thus solves the $n$-step-Life problem with perfect accuracy. We conclude that this minimal neural network architecture can solve the $n$-step-Life problem with a $2n + 1$ layer convolutional neural network with $23n + 2$ weights. In principle, a neural network with an identical architecture should be able to learn a similar solution.

### 3.1 LIFE ARCHITECTURE

We construct a class of architectures to measure how effectively networks of varying sizes solve a hidden-step computational problem. In particular, we employ an architecture similar to the one described in the previous section: an entirely convolutional neural network with $n$ copies of a convolutional layer with $3 \times 3$ filters that feed into a convolutional layer with $1 \times 1$ filters, and finally, a convolutional layer with a single $1 \times 1$ filter and sigmoid activation to decode the output into a Life

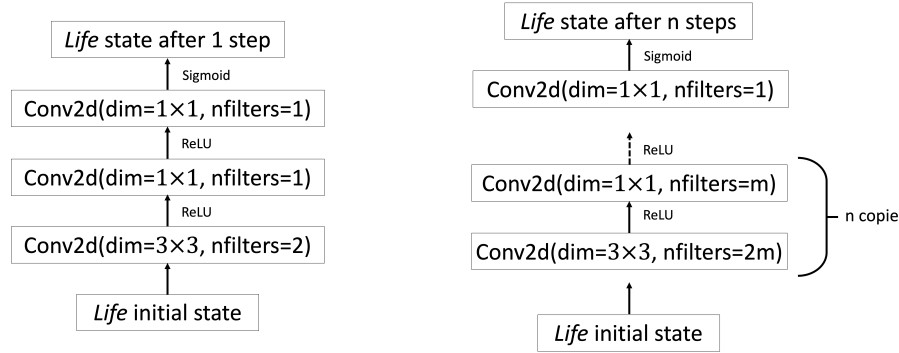

Figure 2: Neural network architecture diagrams for the 1-step minimal model and $\mathcal{L}(n, m)$. On the left, the 1-step minimal architecture consists of an input layer that feeds into a convolutional layer with two $3 \times 3$ filters with ReLU activation, then into a convolutional layer with one $1 \times 1$ filter with ReLU activation, fed into a similar convolutional layer but with sigmoid activation for decoding. On the right, the architecture for $\mathcal{L}(n, m)$ consists of the same as the minimal model, except where the first two hidden layers consist of $2m$ and $m$ filters respectively, and are repeated $n$ times, where $m$ is the factor of overcompleteness (see text).

configuration. When the architecture has $n$ copies of the described layers, we say that it is an $n$-*step architecture*. In the minimal[1] solution, each repeated $3 \times 3$ convolutional layer has two filters and each repeated $1 \times 1$ convolutional layer has one filter. When a similar architecture has $2m$ $3 \times 3$ filters and $m$ $1 \times 1$ filters in each respective repeated layer, we say that the architecture is $m$-*times overcomplete* with respect to the minimal architecture. We let $\mathcal{L}(n, m)$ describe the $n$-step $m$-times overcomplete architecture.

To train instances of each architecture, we initialize the weights randomly from a unit normal distribution. The networks are implemented in Keras (Chollet, 2015) on top of TensorFlow (Abadi et al., 2015) and trained using the Adam optimizer ($\alpha = 0.001, \beta_1 = 0.9, \beta_2 = 0.999$) (Kingma & Ba, 2014) with a binary cross-entropy loss function on the output of the model. Each instance is trained with 1 million randomly generated training examples separated into 100 epochs of 10,000 training examples each, with a batch size of 8. Each training and testing example is generated as follows: first, we uniformly draw a *density* $d$ from $[0, 1]$, and then generate a $32 \times 32$ cell board such that each cell is alive with probability $d$. It is extremely unlikely that the network will ever see the same training example twice. Thus, separating testing data into a testing set and a validation set is unnecessary, since novel data can be generated on the fly. To improve computational efficiency, all networks with the identical parameters are implemented so that they can be trained in parallel using the same randomly generated dataset.

## 3.2 THE DIFFICULTY OF LIFE

To quantify the effectiveness of a given neural network architecture, we measure the probability that a random initialization of the network converges to a solution after being shown one million training examples. Because $\mathcal{L}(n, m)$ can only implement a $3 \times 3$ update rule in each step of computation and is minimal in this sense, for $\mathcal{L}(n, m)$ to solve the $n$-step-Life problem, it must learn the underlying rule. Thus, we consider an instance of $\mathcal{L}(n, m)$ to be successful when it learns the correct underlying rule, and can therefore predict $G^n(x)$ with perfect accuracy for all initial states $x$. Any instance of $\mathcal{L}$ that does not have perfect accuracy did not learn the underlying rule and is thus considered unsuccessful. We wish to determine

$$P[\text{ success of } \mathcal{L}(n, m) \mid n, m ]$$

---

[1]A reviewer helpfully pointed out that an even smaller network can be constructed to solve Life, with a single $3 \times 3$ convolution that counts neighbors, outputs a 0 if there are two neighbors, 1 if there are three neighbors, and -1 otherwise, and then is added to the input and fed through a Heaviside activation function. However, our model is minimal given the constraint that we are using a traditional feedforward CNN with ReLU activation.

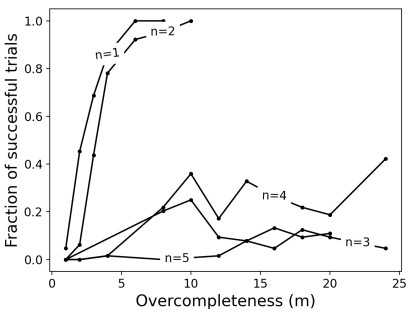

Figure 3: Measured probability that the $m$ times overcomplete $n$-step-Life architecture learns successfully. Each line corresponds with a particular $n$. Each point plots the percentage of 64 instances of $\mathcal{L}(n, m)$ with random initializations sampled from a unit normal distribution that learned the rules of $n$-step-Life after 1 million training examples. We train instances of $\mathcal{L}(n, m)$ for values $1 \leq n \leq 5$ and $1 \leq m \leq 24$, excluding many combinations due to computational constraints. For $n > 1$, none of the instances of $\mathcal{L}$ successfully learned with the minimal($m = 1$) architecture. As $n$ increases, the degree of overcompleteness required for consistent converges increases rapidly.

To accomplish this, we train 64 instances of $\mathcal{L}(n, m)$ for $1 \leq n \leq 5$ and $1 \leq m \leq 24$. We omit certain combinations due to computational limitations. In Figure 3 we plot the percentage of instances of $\mathcal{L}(n, m)$ that successfully learn the $n$-step-Life problem.

We observe that of the minimal ($m = 1$) architectures, only instances for the 1-step-Life problem ($\mathcal{L}(1, 1)$) converged on a solution, with a success rate of approximately 4.7%. Instances of architectures to solve the one and two-step-Life problem had a greater than 50% chance of converging to a solution when the architecture was at least 3 and 4-times overcomplete, respectively. Instances of architectures to solve the $n$-step-Life problem for $n \geq 3$ require an overcompleteness greater than 24, the highest degree of overcompleteness we tested, due to computational constraints. This explosive growth rate suggests that the degree of overcompleteness required for consistent convergence of the $n$-step-Life problem grows quickly with respect to $n$.

Strikingly, for $3 \leq n \leq 5$, we do not observe the hypothesized scaling behavior. Rather, we observe that for high overcompleteness, the architectures for $n = 4$ outperforms $n = 3$, and $n = 5$ performs similarly to $n = 3$. While all three $n$ require many more parameters than the minimal architecture to consistently converge, we would expect that $n = 3$ requires fewer than $n = 4$, which would require fewer than $n = 5$. We have multiple hypotheses: firstly, we may observe this result due to noise or dataset artifacts; secondly, our parameterization of Life may have consistent behavior for all $3 \leq n \leq 5$, which may make the difficulty of learning any $3 \leq n \leq 5$ steps similar.

We plot the loss of the instances of $\mathcal{L}(n, m)$ for $1 \leq n \leq 5$ and $m = 8$ in Figure 4 to illustrate typical rates at which the networks converge to a solution. In addition, we compute the average earliest point of convergence for converged networks of $\mathcal{L}(n, m)$ architecture for $1 \leq n \leq 4$ and $m = 8$ (Figure 4). The earliest point of converge is computed by determining the earliest epoch in which the loss of a convergent network falls below 0.01 to indicate the network has learned a solution to the $n$-step-Life problem. We exclude non-converged networks from this metric.

### 3.3 Weight Perturbations and Learning

To observe the robustness of weight initializations and of learned solutions, we perturb successful weight initializations and solutions of the minimal 1-step-Life architecture $\mathcal{L}(1, 1)$. In particular, we perform two perturbations: the $k$-sign perturbation and the uniform perturbation. The $k$-sign perturbation modifies weights as follows: we select $k$ weights randomly from a uniform distribution. We replace each chosen weight with a weight of the same magnitude but of opposite sign. The uniform perturbation modifies weights by adding a value selected uniformly from the range $[-r, r]$ for a given perturbation magnitude $r$. We choose a weight initialization of a network which converges to a solution to the 1-step-Life problem. We initialize and train instances of the minimal architecture

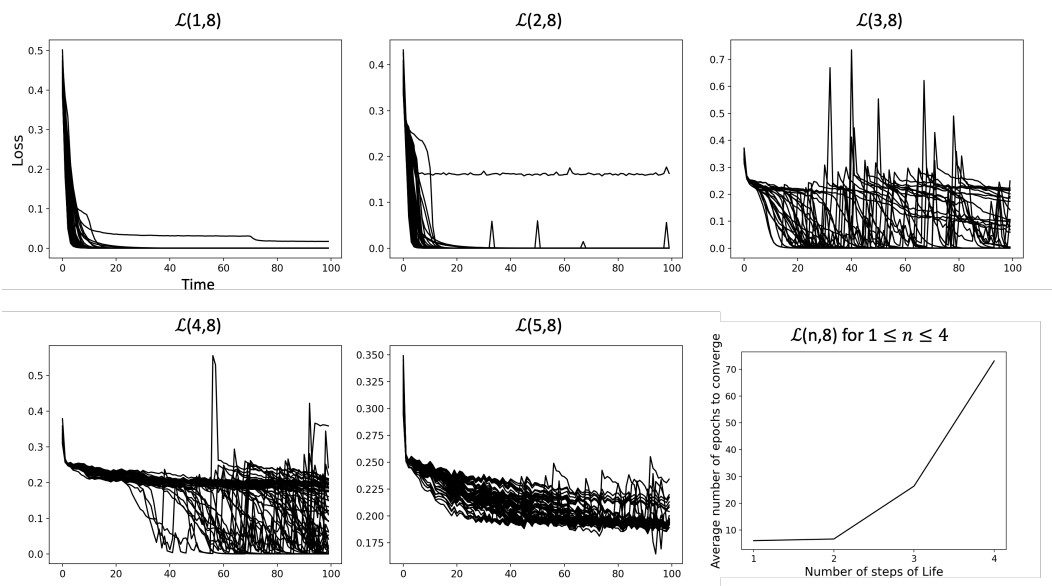

Figure 4: *First five graphs:* Binary cross-entropy loss over the duration of training of the 64 networks trained to solve the $n$-step-Life problem for $1 \leq n \leq 5$. The horizontal axis corresponds to number of epochs of training. For clarity, we omit from the graphs the loss of networks that eventually diverged to a degenerate state where the networks predicted all cells to be dead, regardless of the input Life configuration. These examples converge on a loss that is well over 1. *Last graph:* Average earliest point of convergence of $\mathcal{L}(n, m)$ for $m = 8$ and $1 \leq n \leq 4$. Note that $n = 5$ is excluded because no instances of $\mathcal{L}(5, 8)$ converge. We compute the earliest point of convergence for each network by observing the first epoch where the loss falls below 0.01, indicating that the network has reached a stable 100% accuracy.

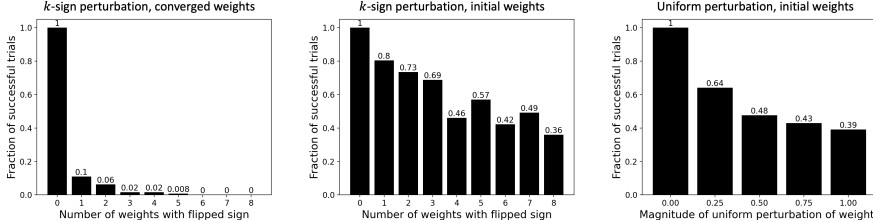

Figure 5: *Left:* Fraction of converged $\mathcal{L}(1, 1)$ networks with weights initialized with a $k$-sign perturbation of a converged solution to the 1-step-Life problem. *Center:* Same as *left* except weights initialized with $k$-sign perturbation of the original initial weights that converged to the solution. *Right:* Same as *center* except weights initialized with a uniform perturbation of the original initial weights. A $k$-sign perturbation of weights is defined as a perturbation in which $k$ weights are chosen randomly from a uniform distribution and replaced with the same magnitude weight with opposite sign. In all cases, 128 networks with the specified weight initialization are trained over 50 epochs, which is well over the number of epochs required for convergence for instances of $\mathcal{L}(1, 1)$. We omit the graph for uniform perturbations of converged weights, as small perturbations have little effect given the magnitude of the converged weights.

with these weights perturbed by $k$-sign perturbations for $1 \leq k \leq 8$ and uniform perturbations for $r \in \{0.25, 0.5, 0.75, 1.0\}$. Similarly, we initialize and train instances of the minimal architecture with the described $k$-sign and uniform perturbations of the converged solution of this network. For each perturbation type, we train 128 instances.

We plot the fraction of successful networks for each perturbation type in Figure 5. Notably, a 1-sign perturbation of the original initial weights of the successful network causes the network to fail to

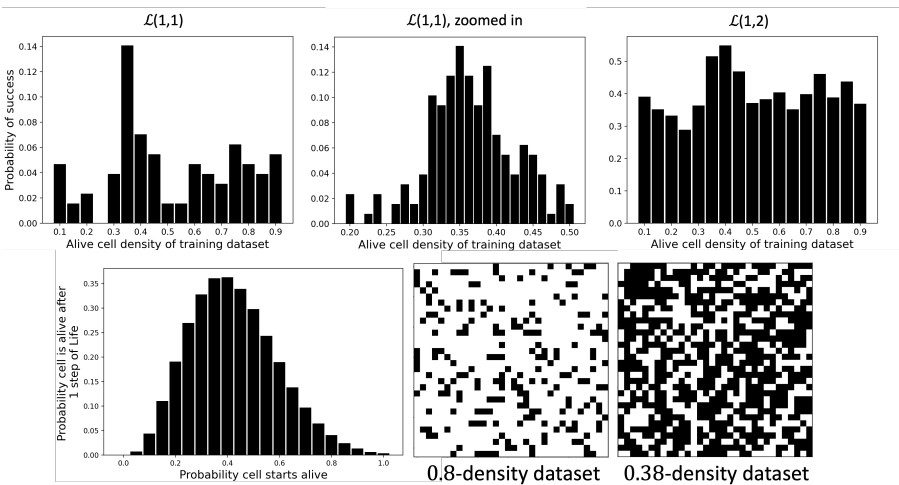

Figure 6: *Top:* Fraction of minimal networks that converge to a solution to the 1-step-Life problem (*left and center*) and 2-times overcomplete networks (*right*) when trained with datasets of a given $d$-density dataset. The *left* and *center* graphs refer to the same training configurations, however, the *left* graph includes the rate of convergence for datasets with $d$ between $0.1$ and $0.9$ with $0.05$ intervals while the *center* graph has data for $d$ between $0.2$ and $0.5$ with $0.0125$ intervals.

*Bottom: Left:* the probability that an arbitrary cell is alive after 1 step of Life of an instance of a $32 \times 32$ cell $d$-density dataset, given $d$. This curve peaks at approximately $d \approx 0.38$. *Right:* two examples from the generated datasets ($0.8$-density and $0.38$-density). We generate a $d$-density dataset by choosing $32 \times 32$ cell grids as training examples where cells are alive with probability $d$.

converge approximately 20% of the time, and only 4–6 sign perturbations are required to drop the success rate below 50%. This suggests that for minimal networks, the weight initialization is sensitive to perturbations. This is not unique to sign perturbations. Even a relatively small uniform perturbation of $0.25$ magnitude (where weights are changed by $0.125$ in expectation) causes the tested networks to fail to learn approximately 36% of the time. Finally, we observe that even a 1-sign perturbation of an already converged solution causes approximately 90% of models to fail to learn, suggesting that converged solutions are very sensitive to sign perturbations. Furthermore, since typical weights are small in the converged solution (weights have a mean of approximately $0.270$ and standard deviation of approximately $-2.17$), sign perturbations do not represent large-magnitude changes.

### 3.4 An Optimal Training Dataset

Many deep learning systems are restricted by the dataset which is available for training. We examine how a class of training datasets affects the success rate of near-minimal networks learning the 1-step-Life problem. In particular, we construct a class of training datasets: the *d-density dataset*, a $32 \times 32$-cell dataset in which cells are chosen independently to be alive with probability $d$. Note that the dataset described in Section 3.1 is a generalization of this class of datasets in which $d$ is chosen uniformly, which we call the *uniform-density* dataset. We show examples of these datasets in Figure 6.

We train 128 instances of $\mathcal{L}(1, 2)$ on the $d$-density datasets for $0.1 \leq d \leq 0.9$ with intervals of $0.05$. We train $\mathcal{L}(1, 1)$ on the same datasets, and in addition, on $d$-density datasets for $0.2 \leq d \leq 0.5$ with intervals of $0.0125$. Surprisingly, we find a sharp spike in probability of success for values of $d$ between approximately $0.3$ and $0.4$. When $d = 0.35$, we observe a 14% success rate for the minimal model, a strikingly high success rate considering that it is approximately double the success rate of the $0.4$-density dataset and triple the success rate of the $0.3$-density dataset (Figure 6). The same result, though less exaggerated, appears for $\mathcal{L}(1, 2)$, where the performance increases drastically for $d = 0.35$ and $d = 0.4$ (Figure 6). The tiny range in which performance increases significantly

suggests that there is likely a critical value $d_0$ such that the $d_0$-density dataset is in this sense optimal. We hypothesize that the value of $d_0$ coincides with the peak of the graph shown in Figure 6, which plots the probability that a cell is alive after one step of Life given that the initial configuration of Life is drawn from a $d$-density dataset, for a given $d$. This would place $d_0 \approx 0.38$. We predict that an optimal dataset must satisfy a condition in which the probability of observing each possible $3 \times 3$ local configuration of Life reaches an equilibrium that allows the computed average gradient of the weights with respect to the loss function and the training examples to direct the weights quickly to a solution. For example, for very small $d$, we expect most cells to be dead after one step of Life given an initial state sampled from the $d$-density dataset. Thus, in this case, the computed average gradient of the weights will tend to drive the network towards a solution which predicts most cells to be dead. The density which maximizes the probability that a cell will be on after one step of Life will maximize the occurrence of cells with exactly three neighbors and alive cells with exactly two neighbors, any instance of which will increase the number of cells that are alive in the next step of Life. We hypothesize that frequent observation of these configurations are critical for a near-minimal network to solve the $n$-step-Life problem.

## 4 Discussion

In this paper, we present four primary results:

1. The minimal network trained with gradient descent rarely converges on a solution to Life.
2. As the complexity of the problem increases, the necessary network size to consistently learn a solution grows rapidly.
3. Ideal initial network parameters are highly sensitive to perturbations.
4. Gradient descent is highly sensitive to distribution parameters of the dataset.

This result is consistent with the lottery ticket hypothesis, which proposes that neural networks converge due to lucky subnetworks that occur due to random initialization which have amenable initial weights for learning the particular task. Thus, it would be unlikely for a near-minimal network to be initialized with weights that are perfect for learning. We characterize this likelihood for networks that model the Game of Life, and in addition show that lucky initialization parameters are highly sensitive to perturbations or non-ideal dataset distributions.

While Conway's Game of Life itself is a toy problem and has few direct applications, the results we report here have implications for similar tasks in which a neural network is trained to predict an outcome which requires the network to follow a set of local rules with multiple hidden steps. Examples of such problems include but are not limited to machine-learning based logic or math solvers, weather and fluid dynamics simulations, and logical deduction in language or image processing. In these instances, without enormously overcomplete networks, gradient descent based optimization methods may not suffice to learn solutions to these problems. Furthermore, such a result may generalize to problems that do not explicitly involve local hidden step processes, such as classification of images and audio, and virtually every other application of machine learning. In addition, significant effort has gone into developing faster and smaller networks with similar performance to their larger counterparts. Our result suggests that these smaller networks may necessarily require alternative training methods, or methods to identify optimal weight initializations.

The highly specific ideal dataset distribution for learning the Game of Life may be specific to the particular networks we train or the problem itself. However, other neural networks, especially small networks, may suffer from similar problems. The dataset parameters may need to be tuned near perfectly in order to maximize the learning potential of neural networks.

In conclusion, we find that networks of the $\mathcal{L}$ architecture that are trained to predict the configuration of Life after $n$ steps given an arbitrary initial configuration require a degree of overcompleteness that scales quickly with $n$ in order to consistently learn the rules of Life. Similarly, we show that weight initializations and converged solutions are extremely sensitive to small perturbations. Finally, we find that these networks are dependent on very strict conditions of the dataset distribution in order to observe a significant increase in success probability. These observations are consistent with the predictions of the lottery ticket hypothesis, and have important consequences in the field.

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

## A    WEIGHTS FOR MINIMAL ARCHITECTURE

We describe weights that solve Life for the minimal architecture $\mathcal{L}(1,1)$.

The first layer has two $3 \times 3$ convolutional filters, each with bias, described as follows:

$$W_{1,1} = \begin{pmatrix} 1 & 1 & 1 \\ 1 & 1/10 & 1 \\ 1 & 1 & 1 \end{pmatrix}$$

$$b_{1,1} = -3$$

$$W_{1,2} = \begin{pmatrix} 1 & 1 & 1 \\ 1 & 1 & 1 \\ 1 & 1 & 1 \end{pmatrix}$$

$$b_{1,2} = -2$$

where $W_{1,1}$ and $W_{1,2}$ describes the weights of the first and second convolutional filters, respectively, and $b_{1,1}$ and $b_{1,2}$ similarly describes the bias. Each output is fed through a ReLU function.

The second layer has a single $1 \times 1$ filter which combines the output of the two filters from the previous layer.

$$W_{2,1} = (-10) \oplus (1)$$

where the first component corresponds with the first-layer output of the first filter and the second component corresponds with the output of the second filter. Each output is fed through a ReLU function.

We add one more layer, as a convenience for learning:

$$W_3 = (2s) \, , b_3 = -s$$

for an arbitrary large $s$, and in practice, 20. The output is then fed through a sigmoid function.

Thus, the entire architecture, given a Life input $x$, has that $N(x)$ computes the next step, where $N$ is defined as:

$$
\begin{aligned}
N_1 &= \text{ReLU}((x \circledast W_{1,1} + b_{1,1}) \oplus (x \circledast W_{1,2} + b_{1,2})) \\
N_2 &= \text{ReLU}(N_1 \circledast (W_{2,1} \oplus W_{2,2})) \\
N &= \text{Sigmoid}(N_2 \circledast W_3 + b_3)
\end{aligned}
$$

