# OpenReview forum: "It's Hard for Neural Networks to Learn the Game of Life"
_ICLR.cc/2021/Conference — Reject_

### Official Review · AnonReviewer3 · 2020-10-22
**Interesting lab setting for exploring harder learning problems**

**Rating:** 6
**Confidence:** 4

**Review:**

This work presents a numerical study of how well can deep learning frameworks learn the underlying rules of a discrete dynamical system when the training set is composed of pairs of configurations separated by some time interval. Specifically, they focus on the Game of Life and study the success rates of learning as a function of the time-interval, perturbations to the weights, and the over-parameterization. The latter is defined as the number of parameters divided by that of a certain ideal DNN that the authors hand-crafted.

They find that over-parametrization is useful when the time interval is small but, at least within their numerical reach, a time interval of 3+ Game of Life steps seems to be always hard to learn. In addition, they find that when the success rate is small taking a converged instance, flipping a weight, and continuing training has a drastic harmful effect on the success rate.

I believe the above two numerical observations (hardness of 3+ steps and the drastic effect of a sign-flip on a converged instance) are interesting and may stimulate further numerical and analytical study. On the other hand, the lack of theory and the fact that these observations are shown in just one setting diminishes the potential impact of this paper.

I suggest several specific points which the authors may wish to try and improve:

1. It would be nice to see whether milder conditions of success, such as getting a sizable portion of the inputs exactly right, are met more often. In a related manner, I wonder if the authors can explain the second observation based on the fact that flipping a single weight, may result in a network which is perfectly accurate for some finite fractions of the inputs, whereas making this weight cross zero back to its original values must pass through a point which ruins the performance on this good finite fraction.

2. The results and approach of this work have similarities with https://openreview.net/forum?id=BJGWO9k0Z, where the Game of Life is traded for a cellular automaton that percolates through a graph. I think the authors should relate both results as they may strengthen one another.

3. It is unfortunate that the transition from success to failure is so sharp with the steps and makes it hard to deduce whether overparameterization improves performance for 3+ steps. Can the authors improve the quality of the numerics there? Perhaps by using more statistics or adopting a more relaxed criterion for success (such as in 1. for instance)? Perhaps by reducing the training-set size? In this aspect, the authors' definition of overparameterization is with respect to the handcrafted DNN. Some define overparameterization w.r.t. the size of the training-set,  as often happens in practical applications of DNNs. Can the authors comment what happens when they enter this latter over-parametrization regime?

4. The authors argue that the success rate is highly sensitive to perturbations in the *initial state*. I see such sensitivity when sign-flipping the converged weights. However, I don't see a striking sensitivity when perturbing initial states (that is, one which is beyond that expected by the fact that the network rarely converges). Can the authors elaborate more on what they mean here? Is there some new information in the study of perturbation to the initial state which is not obvious from the success rate they find?

---

### Official Review · AnonReviewer4 · 2020-10-26
**Interesting work, yet some methodological considerations are warranted**

**Rating:** 5
**Confidence:** 3

**Review:**

This paper uses Conway's Game of Life (GoL) as a testbed for evaluating the learning dynamics of neural networks as a function of their architecture or the random initialization.  In particular, they look at predicting the state of GoL $n$ steps into the future given an arbitrary random initial state, using a multilayer CNN that is shown to be capable of encoding a correct solution to this problem. This network can be $m$-times over-parametrized by adding $m$-times more filters to the convolutional layers. The authors show that in practice, this network rarely finds solutions, especially when $m$ is small or when $3 \leq n \leq 5$, for which it is necessary to use a much larger number of filters (for instance, when $n=1$, the system with just an additional pair of filters finds more reliably solutions than when $n=4$ but 10 times more filters are provided). Finally, the authors show that the systems' convergence probability is sensitive to the initial weights.

In this reviewer's opinion, Conway's Game of Life indeed constitutes an interesting testbed to study the capabilities of neural networks thanks to its rich dynamics, which have been shown to be Turing-complete. The authors further claim that they expect that this study might extend to other architectures and problems. Furthermore, they extrapolate their results to support the claim that neural networks require a large number of parameters to learn any domain, and small networks are sensitive to perturbations.

+ (+) One the main positive aspects of this paper are the experimental results on characterizing the probability of convergence of a network for a given the $n$-step learning problem, with an $m$-times over-parametrized network.

+ (+) They are also very interesting the results that characterize the probability of success of training the network as a function of the initial density, and show that the optimal training density is consistent with a well-known attracting point of $\rho=0.37$ (see, e.g. https://arxiv.org/pdf/1407.1006.pdf).

+ (+) The authors also do some experiments characterizing the basin of attraction of solutions by perturbing the weights of converged networks and of their random initializations before training, showing that the networks are sensitive to the studied perturbations. (See also http://papers.nips.cc/paper/395-back-propagation-is-sensitive-to-initial-conditions.pdf for a related study that is very relevant to this work).

- (-) On the other hand, I believe there might be some issues with how the $n$-step learning problem is analyzed. In particular, it is worth noting that the authors decided to model this by replicating the convolutional blocks $n$ times. Yet, in doing so, they conflate two different issues: On the on side, there is the expected problem of learning to predict $n$ steps in the future. Yet, on the other hand, there is the fact that each of these sub-networks need to learn the transition function independently from each other! Thus, the problem becomes exponentially harder with $n$ (which might in part explain the very poor performances for $n > 2$. To differentiate between these two aspects, it would have been extremely beneficial to have a setup in which the weights for these blocks are _shared_.

- (-) Given that the role of the initialization is greatly emphasized, it surprised me that just a single (and not much motivated) initialization scheme was tried, namely, the unit ball. Similarly, the strong reliance on Adam with default parameters as the only optimization algorithm would require some justification.

- (-) The claim that the number of parameters of the network needed to learn a correct solution grows with the complexity cannot be fully substantiated for the reason that the the scaling breaks for $n > 2$. On one hand, the authors show that more parameters facilitate the learning for $n=1$ or $2$. On the other hand, they show that for $n>2$, if you want to have a shot at learning the problem, you better use a larger network. Yet, there is no consistent improvement, nor clear relation, and thus the claims and conclusions need to be more subtle.

- (-) The title is catchy, but inaccurate. After all, the authors show that neural networks can learn the Game of Life quite easily. You just need more parameters and, say, a single evolution step, but that's exactly what both neural networks and game of life are all about (over-parametrized models and a system with markovian rules)


**Questions for the authors**

- What are you trying to show with Figure 4? It doesn't seem to drive any specific point in the main text.

- "We have multiple hypotheses: firstly, we may observe this result due to noise or dataset artifacts; secondly, our parameterization of Life may have consistent behavior for all 3 ≤ n ≤ 5, which may make the difficulty of learning any 3 ≤ n ≤ 5 steps similar." -> Did you do anything in order to find support for either of these two?

- $\mathcal{L}$ is for "Life"? You might want to mention that, because we see this symbol so many times associated with Loss that it takes some time getting used to that.

**Related references**

The authors might consider referring to https://arxiv.org/pdf/2002.03896.pdf, https://arxiv.org/pdf/1903.12508.pdf and https://arxiv.org/pdf/1908.06663.pdf for other work learning on Life or Life-like environments.

---

### Official Review · AnonReviewer1 · 2020-10-26
**Interesting idea but a lot to improve in the execution**

**Rating:** 3
**Confidence:** 5

**Review:**

The authors study the problem of learning the rule for the game of life with a convolutional network. This is indeed an interesting problem that can be a great benchmark for studying the lottery ticket hypothesis and related training issues - kudos to the authors for this great idea. Unluckily, the methods and results of the paper do not stand up to the idea. While the game of life is an interesting problem, the methods employed to study it must be comparable to modern deep networks so that the results can be meaningfully interpreted in the broader context. The authors study convolutional models that are to learn multi-step processes, but they use neither residual connections in their models nor do they employ normalization or an appropriate multi-layer initialization scheme like fixup. There is also no study of the optimizer - Adam is used with the default Keras hyperparameters and no ablations are performed, even though other works find crucial benefits in tuning them - and also the epsilon parameter that's not even mentioned by the authors. Therefore, while we find the idea very promising, we need to recommend rejection of this paper until the methods are improved.

---

### Official Review · AnonReviewer2 · 2020-10-29
**Interesting experiments but questionable significance**

**Rating:** 5
**Confidence:** 4

**Review:**

# Paper Summary

This paper experimentally studies the problem of training a neural network to learn the rules of Conway's Game of Life from examples using gradient descent. The motivation is that since we can know and construct the minimal networks that can implement the Game of Life, it is interesting to study whether and under what conditions conventional gradient-based training of networks succeeds on this problem. The experiments show that training rarely works for the minimal network architectures, and substantial growth in network size is necessary to make training more reliable. Training is also very sensitive to initial network parameters and data distribution characteristics. Based on these results, the paper suggests that the Game of Life may serve as an interesting and useful test bed for studying general problems of neural network training.

# Strengths

The paper is a well executed and clear experimental study on the problem of learning the Game of Life using neural networks. The experiments are based on simple genuine curiosity, and I found the results interesting (they are not predictable). There is potential for such experiments to lead to interesting ideas that we do not foresee yet.

# Weaknesses

While the experimental results are interesting, I found the contributions to the big picture issues of training neural networks to be unclear due to a couple of main reasons.

A main connection made in the paper is to the lottery ticket hypothesis. The authors claim that since networks much larger than the minimal architecture are needed for reliable training, the findings support the lottery ticket hypothesis. I think there are definitely some relations because initialization plays a critical role here, the relevance of lottery ticket hypothesis is not very clear because of the differences in behavior for $n \in \{3, 4, 5\}$ and $n \in \{1, 2\}$ in Fig. 3. If the main issue was finding lucky subnetwork initializations, wouldn't we expect more consistent trends as Overcompleteness increases for all $n$?

Moreover, I think that there is a very relevant line of prior work along similar lines that needs to be acknowledged and discussed here. [A] and [B] showed that certain synthetic problems that had been presented in prior work [C] as being very difficult for recurrent networks trained by gradient descent (due to long time lags) could actually be easily solved using random weight guessing. The key message was not that weight guessing is a good learning method, but that those benchmarks were not the right ones to focus on when looking for generalizable insights relevant to practical problems. This work is directly relevant here since it is also about the use of synthetic benchmarks to guide progress in learning algorithms.

[A] Schmidhuber, J. and Hochreiter, S., 1996. Guessing can outperform many long time lag algorithms. (appears to be a TR for [B]) http://citeseerx.ist.psu.edu/viewdoc/summary?doi=10.1.1.45.1132

[B] Hochreiter, S. and Schmidhuber, J., 1997. LSTM can solve hard long time lag problems. In Advances in neural information processing systems (pp. 473-479). http://papers.nips.cc/paper/1215-lstm-can-solve-hard-long-time-lag-problems.pdf

[C] Bengio, Y., Simard, P. and Frasconi, P., 1994. Learning long-term dependencies with gradient descent is difficult. IEEE transactions on neural networks, 5(2), pp.157-166.

# Review Summary

The experimental study is very interesting, but I am hesitant to accept this paper due to the weaknesses mentioned above. I would be happy to re-consider my decision based on authors' responses to my concerns. In particular, how do the authors frame their work in the context of prior work that I mentioned? Is there a specific reason to believe that Life is not a "trivial" problem in the sense defined by [A,B], or is it simply an open question at this point? Relatedly, have they tried random weight guessing or variants for training the minimal architecture? Can the authors make any predictions relevant to the lottery ticket hypothesis based on their experimental results? In other words, how do these results indicate that if a method improves reliability of training on Life, we can also expect it to help on other practical problems?

---

### Decision · Program_Chairs · 2021-01-07
**Final Decision**

**Decision:**

Reject

**Comment:**

Unfortunately, the authors did not submit a response during the rebuttal phase.